# A Live Attenuated H1N1 Influenza Vaccine Based on the Mutated M Gene

**DOI:** 10.3390/vaccines12070725

**Published:** 2024-06-29

**Authors:** Yinglei Yi, Hongbo Zhang, Youcai An, Ze Chen

**Affiliations:** 1Department of Basic Research, Ab & B Bio-Tech Co., Ltd. JS, Taizhou 225300, China; anyoucai@abbbio.com.cn; 2Shanghai Institute of Biological Products, Shanghai 200052, China; yiyinglei@163.com

**Keywords:** influenza virus, live attenuated vaccine, M gene, cross-protection

## Abstract

The influenza vaccines currently approved for clinical use mainly include inactivated influenza virus vaccines and live attenuated influenza vaccines (LAIVs). LAIVs have multiple advantages, such as ease of use and strong immunogenicity, and can provide cross-protection. In this study, the M gene of the PR8 virus was mutated as follows (G11T, C79G, G82C, C85G, and C1016A), and a live attenuated influenza virus containing the mutated M gene was rescued and obtained using reverse genetic technology as a vaccine candidate. The replication ability of the rescued virus was significantly weakened in both MDCK cells and mice with attenuated virulence. Studies on immunogenicity found that 1000 TCID_50_ of mutated PR8 (mPR8) can prime strong humoral and cellular immune responses. Single-dose immunization of 1000 TCID_50_ mPR8 was not only able to counter the challenge of the homologous PR8 virus but also provided cross-protection against the heterologous H9N2 virus.

## 1. Introduction

Influenza has plagued mankind for a long time, with four worldwide influenza pandemics in the last century alone. Influenza is an acute respiratory disease caused by the influenza virus, which is highly contagious and spreads rapidly [1]. In early April 2009, a novel influenza A (H1N1) virus emerged in Mexico and the United States and spread rapidly worldwide, causing the first influenza pandemic of the 21st century. COVID-19, caused by severe acute respiratory syndrome coronavirus 2 (SARS-CoV-2), which broke out at the end of 2019, has continued to cause millions of deaths worldwide and poses a serious threat to human public health [2]. Coronavirus disease (COVID-19) and influenza are respiratory infectious diseases with many similar clinical symptoms. People who have both influenza and COVID-19 may have a more severe condition than those who have influenza or COVID-19 alone, which exacerbates the patients’ condition, making treatment more complex and difficult [3]. In addition, some COVID-19 patients may be affected by post-COVID-19 conditions (also known as long COVID). According to an investigation report by the CDC in the United States, the influenza epidemic in 2022–2023 was more serious than that in the past two years during the COVID-19 pandemic. The 2022–2023 influenza epidemic in the UK has been more severe than in the previous two years [4].

Vaccination has always been the most effective measure for preventing influenza, and high-risk groups must pay special attention to it. Currently, the most widely used influenza vaccines are the traditional inactivated trivalent and quadrivalent vaccines; although they have a good safety and efficacy profile, the production process is complex, and the production cycle is time-consuming, making it difficult to provide a sufficiently effective vaccine for use in the early stages of influenza virus epidemics and pandemics [5]. Compared with traditional inactivated influenza vaccines (IIVs), a live attenuated influenza virus vaccine (LAIV) has numerous advantages, as it can simulate natural infection through the replication of the virus in the human upper respiratory tract after vaccination, which can induce a more comprehensive immune response in the body and enable the subject to obtain a more extensive and effective humoral and cellular immune response compared with the inactivated vaccine [6]. In addition, LAIVs can induce a strong cross-immune response, providing strong cross-protection against antigenic drifted viruses of the homosubtypic virus, as well as protection against heterosubtypic viruses [7]. Moreover, live attenuated vaccines may have beneficial non-specific effects in preventing vaccine-unrelated infections [8].

The M gene of the influenza virus encodes both the matrix 1 (M1) and matrix 2 (M2) proteins through alternative splicing. M1 is the most abundant protein in influenza virus particles and is involved in the composition of the viral capsid. It can also bind to the RNA of the virus, interact with the RNP complex, and play an important role in the replication and infection cycles of the virus [9]. The M2 protein, which functions as an ion (mainly Na+) channel and regulates the pH inside the viral particle, can affect the conformation of the HA protein, thereby affecting virus budding and infection [10]. Some studies have shown that mutating the non-coding region of the M gene can reduce its mRNA levels [11]. Additional studies have shown that mutations introduced at the M gene cleavage site can impair the shearing efficiency of influenza viruses at the mRNA level and reduce the expression of the M1 and M2 proteins, thus weakening the replication of the virus [11,12].

In this study, we generated a live attenuated viral strain based on a novel mutated M gene. Serial passages on MDCK cells confirmed its genetic stability. In addition, its pathogenicity in mice was significantly reduced. Next, we evaluated the immunogenicity of the obtained mutant virus as a candidate strain for an LAIV in BALB/c mice, assayed relevant indices of humoral and cellular immune responses, and evaluated its protective effects against homotypic and heterosubtypic influenza viruses.

## 2. Materials and Methods

### 2.1. The Viruses, Animals, and Cells

The viruses used in this study included the murine adapted strains A/PR/8/34 (H1N1) and A/Chicken/Jiangsu/7/2002 (H9N2), which were stored at −80 °C. Specific pathogen-free (SPF) female BALB/c mice (6–8 weeks old) were purchased from the Shanghai Experimental Animal Center. The mice were housed in the Animal Resource Center of the Shanghai Institute of Biological Products, where they were maintained under a 12 h light–dark cycle and in quiet conditions. The room temperature was kept at 26 °C, and the mice were kept in specific pathogen-free conditions. HEK293T and MDCK cells were cultured in MEM (Gibco-BRL) containing 10% fetal calf serum (Invitrogen, Life Technologies, New York, NY, USA).

### 2.2. Plasmid Construction and Virus Rescue

The nucleotides of the M gene from the PR8 virus were mutated as follows, G11T, C79G, G82C, C85G, and C1016A, using overlap PCR. The PCR products were first digested with BsmBI and then cloned into the pHW2000 plasmid. The influenza virus containing a mutated M gene, named mPR8, was rescued using reverse genetic techniques, as described previously [13]. Briefly, the eight plasmids included pHW2000-M(m), pHW2000-PB2, pHW2000-PB1, pHW2000-PA, pHW2000-HA, pHW2000-NP, pHW2000-NA, and phW2000-NS, and 1 μg of each was mixed with 18 μL of the transfection reagent Lipofectamine2000 (Invitrogen), incubated at room temperature for 30 min, and then added to the 60~80% confluent 293T cells in a 6-well plate. Six hours later, the transfection mixture was removed, and Opti-MEM (Gibco-BRL) was added. After 72 h of incubation, the cells and supernatant were harvested and inoculated into 10-day-old SPF chicken embryos. After 2 days of culture at 37 °C, the allantoic fluid of the chicken embryos was harvested and verified using hemagglutination testing and sequencing. After verification, the allantoic fluid was aliquoted and frozen at −80 °C for use as a candidate vaccine strain.

### 2.3. Character of the Rescued Virus

The replication kinetics of mPR8 were investigated in MDCK cells. As described previously [14], the MDCK cells were infected with mPR8 at a multiplicity of infection (MOI) of 0.001, and cell culture supernatant was harvested every 24 h after infection until the end of the 96th hour. The viral titer was determined using the TCID_50_ assay according to the Reed–Muench method.

### 2.4. Pathogenicity Determination in Mice

To determine the pathogenicity of the rescued mPR8 virus, 10 mice per group were anesthetized by intraperitoneal injection of sodium pentobarbital, followed by intranasal inoculation of 20 μL of mPR8 virus fluid at different doses (10 TCID_50_, 100 TCID_50_, 1000 TCID_50_), and the morbidity and survival rate of the mice was observed and recorded for 14 days after infection, with the parental PR8 virus serving as the control. Their body weight was monitored daily, and mice that lost ≥ 25% of their initial body weight were euthanized and defined as dead.

### 2.5. Antibody Assay

Three mice per group were vaccinated with different doses (10 TCID_50_, 100 TCID_50_, 1000 TCID_50_) of the mPR8 virus at a volume of 20 μL via the intranasal route, and mice inoculated with the same volume of PBS were used as the negative control. Twenty-one days after immunization, tail vein blood was collected, and serum was isolated for an antibody titer by ELISA. Virus-specific IgG, IgG1, and IgG2a in the serum and IgA in the nasal lavage fluid were determined according to a previously described method [15,16].

### 2.6. ELISPOT Assay

As described previously, three mice per group were vaccinated with different doses (10 TCID_50_, 100 TCID_50_, 1000 TCID_50_) of the mPR8 virus, and a group of mice inoculated with PBS was included as the negative control. Twenty-one days after immunization, splenocytes were isolated from the mice for the IFN-γ ELISPOT assay according to the instructions of the kit (U-CyTech B.V., Utrecht, The Netherlands) [16].

### 2.7. Flow Cytometry Analysis

As described previously, splenocytes were isolated from the mice three weeks after vaccination, and the distribution of the T cell subsets was examined by flow cytometry. The splenocytes were labeled using APC Hamster Anti-Mouse CD3e, FITC Rat Anti-Mouse CD4 (GK1.5), and PE Rat Anti-Mouse CD8a (Becton, Dickinson and Company). BD FACSCalibur (Becton, Dickinson and Company) was used to detect the distribution of various T cell subsets.

### 2.8. Animal Challenging Experiment

To evaluate the protective effect of the rescued virus, the mice were inoculated intranasally with different doses (10 TCID_50_, 100 TCID_50_, 1000 TCID_50_) of the mPR8 virus, while the mice in the control group were inoculated intranasally with the same volume of PBS, with 6 mice per group. Twenty-one days after vaccination, the mice in each group were evenly and randomly divided into two groups and then challenged with 10 LD_50_ of the PR8 or H9N2 influenza virus. Three days after challenge, three mice in each group were randomly selected and euthanized, and their BALF and nasal lavage fluid were collected for virus titer determination using the TCID_50_ assay. The remaining mice (10 mice in each group) were observed daily, and their morbidity and mortality were recorded.

### 2.9. Statistics

The data between experimental groups were analyzed using Student’s *t*-test; a *p*-value less than 0.05 was considered to be significant. Significant differences in survival between the immunization and control groups were determined using Fisher’s exact test. A one-way ANOVA test was used for comparison between the vaccinated groups and the control group.

### 2.10. Ethical Approval

All the animal experiments were conducted with approval from the Animal Care Committee of Shanghai Institute of Biological Products (ethical approval number: SIBP20180210). The ARRIVE guidelines were followed for all the mouse experiments in this study.

## 3. Results

### 3.1. Rescue of mPR8

Overlapping PCR was applied to introduce mutation sites (G11T, C79G, G82C, C85G, and C1016A) into the M gene. The mutated M gene was inserted into the pHW2000 vector after verification by sequencing and was named pHW-M(m). Eight plasmids (pHW-M(m), pHW-PB2, pHW-PB1, pHW-PA, pHW-HA, pHW-NP, pHW-NA, and pHW-NS) were co-transfected into the 293T cells, and the cell supernatants were harvested three days later and inoculated into chicken embryos. The rescued virus was confirmed by sequencing and was named mPR8.

### 3.2. Genetic Stability of the mPR8 Virus

The mPR8 virus strain was serially passaged on the MDCK cells, the 1st, 5th, and 10th generation viruses were harvested, and the viral RNA was extracted and sequenced to validate the nucleotide sequence of the virus. Sequence alignment suggested that the M gene of the mPR8 virus strain did not change from generations 1 to 10, indicating that the rescued strain has good genetic stability.

### 3.3. Characteristics of the mPR8 Virus

To determine the growth kinetics of the mPR8 virus strains, MDCK cells were inoculated with the PR8 or mPR8 viruses at an MOI of 0.001, and the supernatant of the cell culture was collected at 24 h, 48 h, 72 h, and 96 h post-infection (p.i.). The virus titer in the supernatant was determined using the TCID_50_ assay on the MDCK cells. The results showed no significant difference between mPR8 and PR8 at 24 h p.i. At 48 h post-infection, the replication competence of the mPR8 vaccine candidate strain decreased significantly compared to PR8, the donor master strain. The results at 72 and 96 h post-infection were consistent with those at 48 h. These results indicated a significant reduction in the replicative capacity of the mPR8 vaccine candidate strain (Figure 1A).

The mice were intranasally infected with 10^5^ TCID_50_ of the mPR8 or PR8 virus, and after 3 days, the mice were euthanized, and the bronchial alveolar lavage fluid was collected for virus titer determination. The results showed that the mean titer of the mPR8 virus infection group was 10^2.1^ TCID_50_/mL and that of the PR8 virus infection group was 10^5.57^ TCID_50_/mL, indicating that the replication competence of mPR8 in the mice’s respiratory tract decreased >1000-fold relative to the PR8 virus (Figure 1B). Furthermore, the LD_50_ of the mPR8 virus was 10^5.7^ TCID_50_, whereas the LD_50_ of the PR8 virus was 10^3.54^ TCID_50_ (Figure 1C), indicating a significant reduction in the pathogenicity of the mPR8 virus.

The constrained replication of mPR8 on the MDCK cells and in mice indicates that its virulence and ability to cause illness in mice were reduced and it could be used as a live attenuated influenza vaccine candidate.

### 3.4. Humoral Immune Response Induced by mPR8 Immunization

In the humoral immune response studies, grouped BALB/c mice were immunized with 10 TCID_50_, 100 TCID_50_, or 1000 TCID_50_ of the mPR8 candidate strain, and the mice in the control group were inoculated with PBS. Twenty-one days after immunization, the mice were euthanized, and their sera and nasal lavage fluids were collected for antibody assays.

The sera from the 1000 TCID_50_ immunization group showed a high titer of IgG antibodies (2^15.67^). In addition, higher titers of IgA were detected in the nasal lavage fluid of mice immunized with 1000 TCID_50_ of mPR8. In the antibody response assay, the levels of IgG1 and IgG2a were positively correlated with the immunization (Table 1 and Table 2, Appendix A). Analysis of the IgG1 and IgG2a titers showed that mice immunized with the mPR8 strain had a balanced immune response, with no significant Th-biased immune response.

### 3.5. Cell Immune Response Induced by mPR8 Immunization

Mice were immunized with different doses of the mPR8 vaccine candidate strain, and their splenocytes were isolated 21 days later for the cell immune response assay.

The number of interferon-γ-secreting T cells was detected by ELISpot assay. As shown in Figure 2, more IFN-γ-secreting T cells were detected in the 1000 TCID_50_ immuno-group than in the 100 TCID_50_ dose group. However, IFN-γ-secreting T cells were barely detectable in the 10 TCID_50_ dose group, as well as in the control group, which suggests that the number of effector T cells increases with the dose of immunization and correlates positively with the dose of the vaccine. The specific T cell response induced by mPR8 vaccination was further evaluated by examining the distribution of the T cell subsets in the splenocytes. As shown in Figure 3A,B, with an increasing vaccine dose, different subsets of T cells (CD4^+^ T cells and CD8^+^ T cells) also increased significantly, indicating a dose-dependent cellular immune response to the vaccine candidate. Among them, the proportions of CD4^+^ T cell and CD8^+^ T cell subsets in the control group were 3.29% and 1.42%, respectively, while the percentages of CD4^+^ T cell and CD8^+^ T cell subsets were 10.62% and 4.53%, respectively, in the 1000 TCID_50_ immunization group. These results suggest that the mPR8 vaccine candidate can prime a robust T cell immune response and that the level of the cellular immune response is positively correlated with the dose of the vaccination.

### 3.6. mPR8 Vaccination Provides Protection against Homologous Virus Challenging

Thirteen BALB/c mice in each group were intranasally immunized with 10 TCID_50_/mouse, 100 TCID_50_/mouse, or 1000 TCID_50_/mouse of the mPR8 vaccine candidate strain, and the mice in the control group were intranasally inoculated with PBS. Twenty-one days after immunization, the mice were challenged with 10 LD_50_ of the PR8 virus. Three days after PR8 virus infection, three mice in each group were euthanized, and BALF was collected for viral titer determination. The remaining 10 mice in each group were observed for 21 days after challenge, and the survival rate and weight loss of the mice in each group were observed and recorded to evaluate the effectiveness of the candidate vaccine against the homologous influenza virus A/PR/8/34 (H1N1). The results showed that after PR8 virus infection, all the mice survived (100% survival rate) in the 1000 TCID_50_/mouse immune group (Figure 4A); in the 100 TCID_50_ dose group, only 30% of the mice survived, and in the 10 TCID50/mouse dose group, all the mice died on day 8 after challenge. The 1000 TCID_50_/mouse-immunized group showed only slight weight loss and returned to their original body weight by day 9 post-infection, while the mice in the 100 TCID_50_/mouse and 10 TCID_50_/mouse groups all had much more severe weight loss (Figure 4B). The mice in the control group suffered the most severe weight loss and died on day 7 after the challenge (Figure 4A,B).

The virus in the respiratory tract was titered, as shown in Table 3. In the control group, the highest levels of viral replication, 10^7.05^ TCID_50_/mL and 10^4.56^ TCID_50_/mL, were detected in the lungs and nasal turbinates, respectively. The mice in the 1000 TCID_50_/mouse immunization dose group had the lowest titers, of 10^4.17^ TCID_50_/mL and 10^2.5^ TCID_50_/mL, respectively. In the 100 TCID50/mouse immunization dose group, the viral titers in the mice lungs and nasal turbinates were 10^5.44^ TCID_50_/mL and 10^3.73^ TCID_50_/mL, respectively, which were also significantly reduced compared with the PBS control group. In the 10 TCID_50_/mouse immunization dose group, the viral titers in the mice lungs and nasal turbinates were 10^6.57^ TCID_50_/mL and 10^4.28^ TCID_50_/mL, respectively, which were close to the PBS control group.

### 3.7. The mPR8 Candidate Vaccine Strain Protects against Heterosubtypic Virus Challenge

To evaluate whether different doses of mPR8 induced effective protection against a heterosubtypic virus, the mice were immunized with 10 TCID_50_, 100 TCID_50_, and 1000 TCID_50_ of the mPR8 vaccine candidate strain, and the control group was inoculated with PBS, with 13 mice in each group. Twenty-one days after immunization, the mice were intranasally challenged with 10 LD_50_ of the A/Chicken/Jiangsu/07/2002 (H9N2) virus. Three days after challenge, 3 mice in each group were randomly selected to detect the virus titer in their lungs, and the remaining 10 mice in each group were observed for 21 days after challenge. The survival and body weight loss of the mice in each group were monitored daily. As shown in Figure 4C,D, the highest-dose vaccine immune group had a good protective effect against the heterotypic virus challenge, and its protective efficacy was positively correlated with the dose of the vaccine. The results suggested that the mPR8 vaccine candidate could induce a robust immune response, help mice clear the lung virus in the time after being infected with the heterotypic viruses, and achieve protective effects (Table 4). Analysis of the viral titers in the lungs of the mice immunized with different doses of mPR8 showed that the viral titer in the mice was correlated with the dose of the vaccine.

## 4. Discussion

The prophylaxis of influenza remains a global challenge, and novel influenza vaccines are being investigated by researchers in various countries. Live attenuated vaccines based on strategies such as NS gene truncation [17] and M2 cytoplasmic tail deletion are also being developed [18,19]. In addition, codon de-optimization, bivalent chimeric influenza virus vaccines, and other new strategies for LAIVs are being investigated [14,20,21].

Tokiko Watanabe et al. deleted 11 amino acids from the cytoplasmic tail of M2 and altered the cleavage site of HA to generate the H5N1 (A/Vietnam/1203/04) LAIV candidate strain, which can protect mice against lethal doses of homologous (A/Vietnam/1203/04) and heterologous (A/ Indonesia/7/2005) H5N1 viral attack [19]. Mueller et al. used synthetic attenuated virus engineering to redesign most of the coding regions of the PB1, NP, and HA genes of the influenza virus PR8 [21]. The virulence of the codon-pair de-optimized mutant strain was greatly attenuated compared to that of the wild-type strain. Single-dose immunization with this mutant strain protected mice against the wild-type strain challenge.

The 12 nucleotides at the 3′ end and the 13 nucleotides at the 5′ end of the M gene of the influenza virus are highly conserved, and together, they form the promoter of VRNA transcription of the virus [22]. In the influenza virus genome, the 3′-terminal and 5′-terminal conserved nucleotides connect the viral RNA head to the end through base complement pairing to form a stem–loop structure, which plays an important role in the transcription of the influenza virus genome. First, the presence of this stem–ring structure is critical for the influenza virus polymerase to exert its endonuclease activity and obtain the cap structure from the mRNA of host cells. In addition, the stem–loop structure is important for the terminal polyadenylation of viral mRNA. Moreover, published studies have shown that these stem–loop structures play a role in reinforcing viral RdRp (RNA-dependent RNA polymerase) in binding to RNA and after RNA binding. In this study, mutations at the conserved ends (5′ and 3′) of the M gene were introduced to change its working efficiency without destroying the stem–loop structure of the viral genome. In addition, the research results of Paul Digard et al. showed that the conserved codons in the coding region of the influenza A virus genome were closely related to RNA packaging signals. They carried out synonymous mutations of the conserved codons (determining amino acids) in multiple influenza virus genes, such as PB2, PB1, PA, HA, NA, NP, and NS1, resulting in the dampening of virus replication and proliferation to a greater extent. Through the analysis of the conservation of the nucleotide sequence of the M gene, the region of nucleotide position 70–85 is also relatively conserved, and the nucleotide sequence of this conserved region may also play an important role in the packaging of the influenza virus genome during its replication [23]. To investigate the role of this conserved sequence in the M gene in viral replication and proliferation, we introduced three synonymous mutations at positions C79G, G82C, and C85G. Our results showed that this mutation could further reduce the replication and proliferation of the influenza virus (Figure 1A).

The attenuated properties and genetic stability of LAIVs are the primary considerations and the focus of this study. The in vitro results showed that the replication capacity of the mPR8 vaccine candidate strain was limited and significantly reduced after 48 h of infection in the MDCK cells (Figure 1A). The in vivo experimental results showed that the mPR8 candidate vaccine strain had a lower viral titer in the lungs of mice than the PR8 strain (Figure 1B). At the same time, the LD50 of the mPR8 strain was reduced more than 1000-fold (Figure 1C). Taken together, these results indicate that the vaccine had attenuating properties.

The genetic stability of the mPR8 virus strain was evaluated by subjecting it to successive passages in the MDCK cells and then determining the nucleotide sequences of the M genes of different generations of the virus. The results showed that the mPR8 vaccine candidate strain had good genetic stability, and no nucleotide mutations occurred during passaging. These results indicated that the mPR8 vaccine candidate strain had weakened replication competence and good genetic stability.

The effectiveness of the vaccine was evaluated in terms of both the humoral and cellular immune responses elicited after vaccination. Antibodies, as the main components of the humoral immune response, play a very important role in the fight against influenza virus infection. Our experimental results showed that the titer of serum antibodies increased with an increasing immunization dose, and the group immunized with 1000 TCID_50_ had the highest titer of IgG antibodies, which was significantly higher than that in the other groups (Table 1). The IgG1 and IgG2a isotypes were determined using ELISA. The results suggested that the immune response induced by the live attenuated vaccine mPR8 was Th-balanced without significant bias (Table 2).

Natural influenza virus infection has been shown to be more effective than the currently used inactivated vaccines, especially for cross-protection against various viruses. Protection caused by natural infections is associated with the reactivity of IgA antibodies in the respiratory tract. Intranasal immunization with LAIVs elicits immune responses similar to the various immune mechanisms activated by natural infection with wild-type influenza viruses but without causing the typical signs or symptoms associated with illness. The mucosal antibody IgA is the most effective line of defense against influenza virus infection, and we also examined the response levels of mucosal antibodies in different immune dose groups. The results showed that IgA was detectable in both the 1000 TCID_50_ and 100 TCID_50_ immune groups (Table 1).

The immunogenicity of the vaccine was further evaluated by testing the specific induced cellular immunity. The number of effector T cells reflects the level of cellular immunity to a certain extent. IFN-γ is mainly secreted by effector T cells, and we measured the number of splenocytes that could secrete IFN-γ by ELISpot assay. The results showed that the number of IFN-γ-secreting splenocytes increased with increasing immune doses (Figure 2). The percentage of CD4^+^ and CD8^+^ T cells was measured by flow cytometry, and the results were consistent with the ELISpot results (Figure 3). These results suggest that cellular immunity is enhanced with increasing immune doses.

By recording mortality and body weight loss in mice after virus challenge (PR8 or H9N2), the results showed a positive correlation between the protective effectiveness of the vaccine and the dose administered (Figure 4). By comparing the changes in body weight of the mice, we found that the mice challenged with the homologous virus (PR8) recovered their body weight faster in the 1000 TCID_50_ immunization group, returning to their initial body weight in approximately 2 weeks (Figure 4B), whereas the mice challenged with the heterologous virus (H9N2) recovered their body weight slightly slower relatively (Figure 4D). This result suggests that antibodies specifically directed against the PR8 virus and cellular immunity play a more rapid role in the clearance of homologous viruses. In addition, we did not check the viral load or pathological damage in various tissues or perform other analyses. When using mice as animal models to evaluate the protective effect of influenza vaccine challenge, survival, lung virus titers, and weight loss are generally considered to be the most important indicators by influenza vaccine researchers. In future work, we will further test the viral load in various tissues and pathological damages to evaluate the safety of live attenuated virus vaccines.

Due to the large variability in HA and NA of different subtypes of influenza viruses, the HA and NA antibodies produced after immunization are usually not effective in neutralizing heterotypic viruses [21]. The mPR8 vaccine candidate in our study was able to protect against challenge with a lethal dose of the H9N2 virus. However, earlier studies have shown that antibodies to conserved internal proteins of influenza viruses (such as NP or M1) do not play a role in resisting viral challenge [24,25]. In contrast, recently published papers have shown that non-neutralizing antibodies to NP play a role in alleviating mortality and morbidity and in reducing lung viral titers in mice [26]. Other reports have also demonstrated that non-neutralizing antibodies play an important role in virus clearance in the presence of memory T cells, and some studies have suggested that B cells may also exert their effects in clearing the heterosubtypic influenza virus through a variety of mechanisms [27,28]. Xie et al. found that adoptive immunization with antiserum against H1N1 provided partial cross-protection against the H5N1 virus in mice [29].

Immunity against heterologous viruses is mainly provided by T cells and non-neutralizing antibodies, which do not prevent viral infection but could limit viral replication and greatly alleviate disease severity and mortality [27,30]. Therefore, compared with the traditional inactivated vaccines, LAIVs can elicit a more sustained immune response and provide protection against both wild-type viruses of the vaccine strain and viruses of different subtypes [31]. However, some studies have shown that vaccines exert cross-protection, depending on the presence of B cells, and that cross-protection by memory T cells is not very effective in the absence of B cells [32]. Several studies have found that certain epitope-specific CD8+ T cells play an important role in virus clearance and cross-subtype protection [32].

Some studies have found that virus-specific CD4^+^ and CD8^+^ T cells can be detected in the lungs of mice immunized with a live attenuated vaccine on day 4 after heterosubtypic virus challenge [30]. Other studies have found that live attenuated vaccines can activate the Th1 immune response with higher levels of CXCL9 and CXCL11 in the lungs, and higher levels of cytokines associated with T cell and DC recruitment have also been detected in the lungs [31]. Other reports have demonstrated that CXCL9 and CXCL11 chemokines play an important role in the recruitment of T cells to the site of infection during influenza virus infection [33]. In turn, rapid recruitment of T cells accelerates virus clearance and reduces the inflammatory response in the lungs, thereby reducing morbidity and increasing survival [34]. We speculate that the cross-protective effect is likely the result of a synergistic effect of humoral and cellular immunity; however, further in-depth studies are needed to confirm this speculation.

The main objective of this study was to test the concept that the introduction of point mutations within the M gene of the influenza virus could produce a live attenuated influenza vaccine candidate strain. Therefore, in this study, we only conducted relevant studies using cells and animal (mouse) models. The results of this study demonstrate that our assumptions and strategies are feasible. These results are insufficient for the development of a live attenuated influenza vaccine. In future work, we will conduct relevant preclinical and clinical studies to further evaluate the potential of the attenuated vaccine strain obtained in this study as a live-attenuated influenza vaccine. In addition, we plan to test other backbones in future studies.

## 5. Conclusions

The attenuated influenza viral strain obtained through mutation of the M gene of the virus can be used as a candidate vaccine which can activate humoral and cellular immunity in mice and provide effective protection against homologous and heterologous influenza virus challenge.

## Figures and Tables

**Figure 1 vaccines-12-00725-f001:**
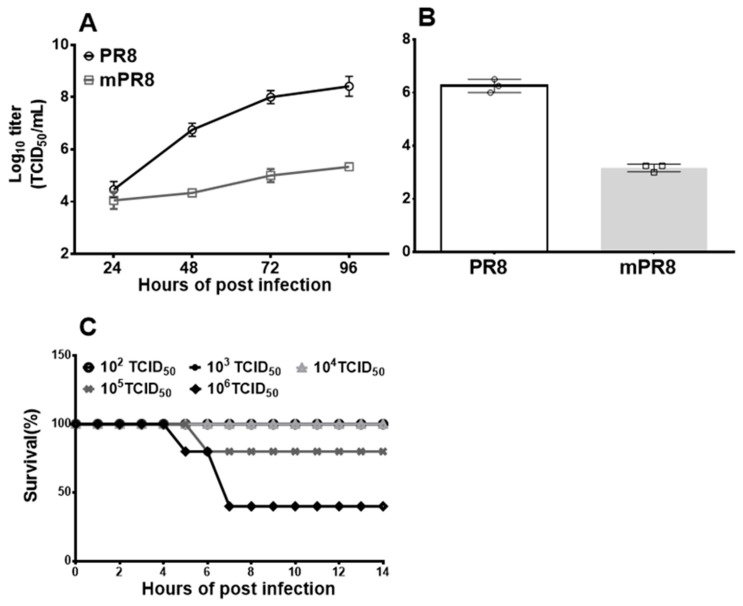
Pathogenicity and virulence of mPR8 virus in vitro and in vivo. The MDCK cells were infected with mPR8 or PR8 at MOI of 0.001 and the cell supernatant collected at 24 h, 48 h, 72 h, 96 h after infection for virus titer determination. The titer was shown in TCID_50_/mL (**A**). The mice were intranasally infected with 20 μL (10^5^ TCID_50_) of mPR8 or PR8 virus, with 3 mice in each group, the mice were euthanized, and their bronchoalveolar lavage fluid was collected for virus titer determination by TCID_50_ assay (**B**). BALB/c mice (*n* = 10 per group) were intranasally infected with 20 μL of different doses of mPR8 virus under anesthesia condition, and the morbidity and mortality were observed and recorded daily (**C**). * *p* < 0.05, represents significant difference between the two groups.

**Figure 2 vaccines-12-00725-f002:**
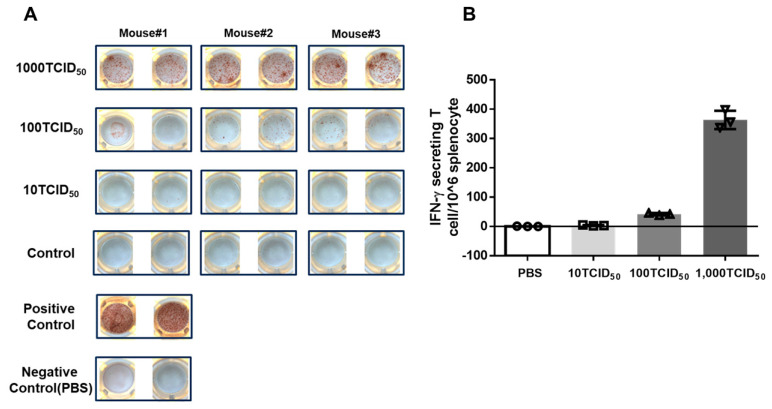
IFN−γ secreting T cells in spleen lymphocytes. BALB/c mice were intranasally immunized with different doses of mPR8 virus as indicated, and the mice in control group were inoculated with the same volume of PBS, with 3 mice per group. Twenty-one days after immunization, the mice were euthanized, and their spleen lymphocytes were isolated. The splenic IFN-γ secreting T cells were determined by ELISpot assay. (**A**) Original pictures of ELISPOT. (**B**) The statistical results of ELISPOT. * *p* < 0.05 means significant difference between the immunization group and PBS control group. Isolated splenocytes cultured in complete culture medium consisting of RPMI 1640 with the addition of 10% FBS, ionomycin (50 ng/mL, InvivoGen), and phorbol myristate acetate (PMA, 10 ng/mL, InvivoGen) were used as positive control.

**Figure 3 vaccines-12-00725-f003:**
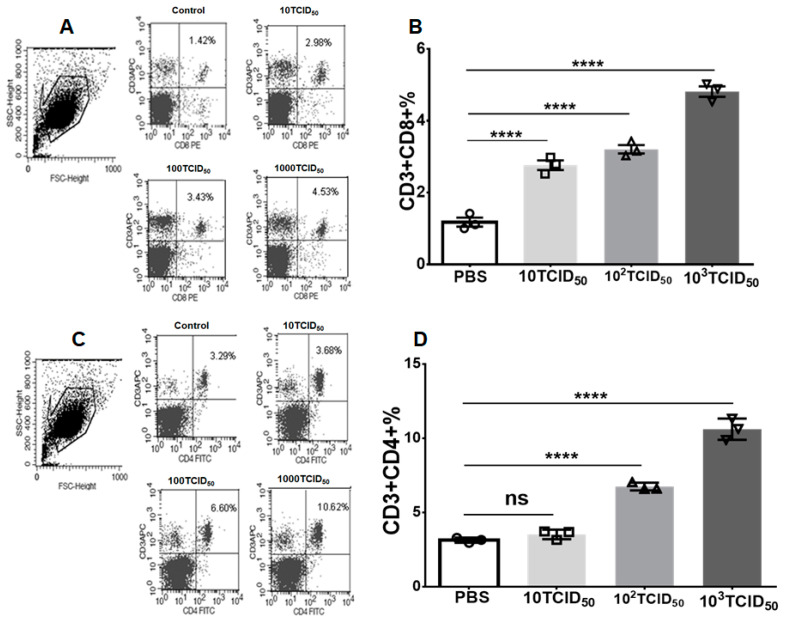
The distribution frequency and percentage of CD3^+^CD8^+^ or CD3^+^CD4^+^ T cells in mPR8 immunized mice. BALB/c mice were intranasally immunized with different doses of mPR8 virus as indicated, and the mice in control group were inoculated with the same volume of PBS, with 3 mice per group. Twenty-one days after immunization, the mice were euthanized, and their spleen lymphocytes were isolated. Distribution frequency and percentage of CD3^+^CD8^+^ T cells (**A**,**B**) and CD3^+^CD4^+^ T cells (**C**,**D**) were analyzed by FACS. A one-way ANOVA test was used for the statistical significance analysis between the vaccinated groups and control group. **** *p* < 0.0001, represents significant difference between the vaccinated group and control group.

**Figure 4 vaccines-12-00725-f004:**
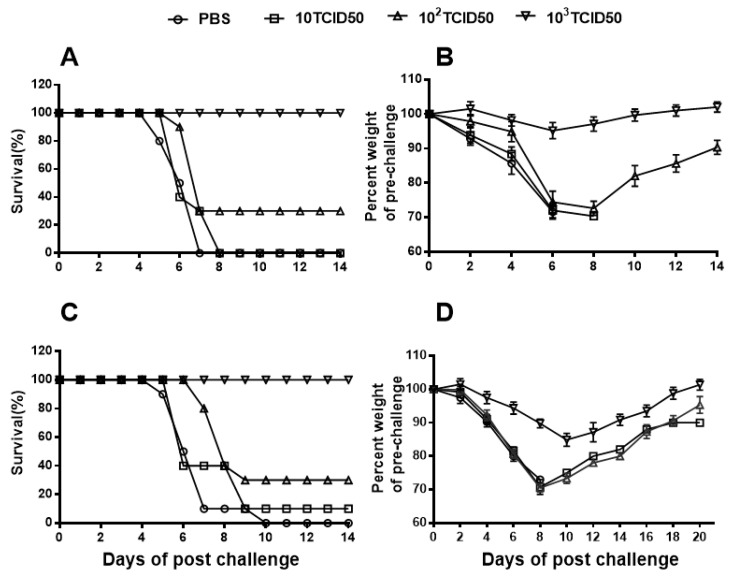
mPR8 protects mice against lethal dose of homologous PR8 virus or heterologous H9N2 virus challenge. BALB/c mice in each group were immunized with 20 μL of different doses (10 TCID_50_/mouse, 100 TCID_50_/mouse, 1000 TCID_50_/mouse) of mPR8 or PBS control and then challenged with 10 LD_50_ of PR8 virus (**A**,**B**) or 10 LD_50_ of H9N2 (A/Chicken/Jiangsu/7/2002) (**C**,**D**) virus after 3 weeks of immunization. The mortality rate (**A**,**C**) and body weight loss (**B**,**D**) were observed and recorded daily following PR8 virus challenge.

**Table 1 vaccines-12-00725-t001:** Antibody responses in mice induced by intranasal administration of various doses of mPR8 vaccine.

Group	Immunogen	Route of Administration	Dose (TCID_50_/Mouse)	Ab Responses (ELISA, 2^n^) ^a^	
				Serum IgG	Nasal Wash IgA
A	mPR8	i.n.	1000	15.67 ± 0.58 ^b,c,d^	6.67 ± 1.15 ^b,c,d^
B	mPR8	i.n.	100	8 ± 1.3	2.67 ± 0.8
C	mPR8	i.n.	10	5 ± 0.3	1.04 ± 0.2
D	PBS	i.n.	-	- ^e^	- ^e^

BALB/c mice were intranasally (i.n.) immunized with different doses of mPR8 virus, as indicated. Mice in the control group were inoculated with the same volume of PBS (three mice per group). Twenty-one days after immunization, serum and nasal lavage fluid were collected for virus-specific IgG and IgA assays, respectively, using ELISA. ^a^ Results of three mice from the same group are shown as mean ± SD; ^b^ *p* < 0.05, significant difference compared with control group; ^c^ *p* < 0.05, significant difference compared with 10 TCID_50_ immune group; ^d^ *p* < 0.05, significant difference compared with 100 TCID_50_ immune group; ^e^ not detected.

**Table 2 vaccines-12-00725-t002:** IgG1 and IgG2a antibody responses in mice induced by intranasal administration of various doses of mPR8 vaccine.

Group	Immunogen	Route of Administration	Dose (TCID_50_/Mouse)	Ab Responses (ELISA, 2^n^) ^a^	
				Serum IgG1	Serum IgG2a
A	mPR8	i.n.	1000	15.33 ± 1.1 ^b,c,d^	16 ± 0.5 ^b,c,d^
B	mPR8	i.n.	100	7.23 ± 0.73	6.67 ± 0.78
C	mPR8	i.n.	10	6.31 ± 0.34	5.1 ± 0.46
D	PBS	i.n.	-	- ^e^	- ^e^

BALB/c mice were intranasally immunized with different doses of mPR8 virus, as indicated, and the mice in the control group were inoculated with the same volume of PBS (3 mice per group). Twenty-one days after immunization, the mice were euthanized, and the serum was harvested for virus-specific IgG1 and IgG2a assays by ELISA. ^a^ Results of three mice from the same group are shown as mean ± SD; ^b^ *p* < 0.05, significant difference compared with control group; ^c^ *p* < 0.05, significant difference compared with 10 TCID_50_ immune group; ^d^ *p* < 0.05, significant difference compared with 100 TCID_50_ immune group; ^e^ not detected.

**Table 3 vaccines-12-00725-t003:** Protection against lethal PR8 virus challenge in mice by intranasal administration of various doses of mPR8 vaccine.

Immunogen	Dose (TCID_50_/Mouse)	Protection against PR8 Virus Challenge		
		Lung Virus Titers ^a^ (log_10_ TCID_50_/mL)	Nasal Wash Virus Titers (log_10_ TCID_50_/mL)	No. of Survivors/ No. Tested
mPR8	1000	4.17 ± 0.60 ^b,c,d^	2.5 ± 0.10 ^b,c,d^	10/10
mPR8	100	5.44 ± 0.18 ^a^	3.73 ± 0.40 ^a^	3/10
mPR8	10	6.57 ± 0.45	4.28 ± 0.37	0/10
PBS	-	7.05 ± 0.47	4.56 ± 0.10	0/10

BALB/c mice were intranasally immunized with different doses of mPR8 virus, as indicated, and the mice in the control group were inoculated with the same volume of PBS (3 mice per group). Twenty-one days after immunization, the mice were challenged with 10 LD_50_/mouse of PR8 virus, and three days after challenge, the mice were euthanized, and the bronchoalveolar lavage fluid and nasal lavage fluid were collected for virus titer determination by TCID_50_ assay. ^a^ Results of three mice from the same group are shown as mean ± SD; ^b^ *p* < 0.05, significant difference compared with the control group; ^c^ *p* < 0.05, significant difference compared with 10 TCID_50_ immune group; ^d^ *p* < 0.05, significant difference compared with 100 TCID_50_ immune group.

**Table 4 vaccines-12-00725-t004:** Protection against lethal H9N2 virus challenge in mice by intranasal administration of various doses of mPR8 vaccine.

Immunogen	Dose (TCID_50_/Mouse)	Protection against H9N2 Virus Challenge		
		Lung Virus Titers (log_10_ TCID_50_/mL) ^a^	Nasal Wash Virus Titers (log_10_ TCID_50_/mL)	No. of Survivors/ No. Tested
mPR8	1000	5.27 ± 0.34 ^b,c^	3.21 ± 0.67 ^b,c,d^	10/10
mPR8	100	6.04 ± 0.67	4.43 ± 0.52	3/10
mPR8	10	6.97 ± 0.48	4.56 ± 0.56	1/10
PBS	-	7.05 ± 0.52	4.78 ± 0.43	0/10

BALB/c mice were intranasally immunized with different doses of mPR8 virus, as indicated, and the mice in the control group were inoculated with the same volume of PBS (3 mice per group). Twenty-one days after immunization, the mice were challenged with 10 LD_50_/mouse of H9N2 virus, and three days after challenge, the mice were euthanized, and their bronchoalveolar lavage fluid and nasal lavage fluid were collected for virus titer determination by TCID_50_ assay. ^a^ Results of three mice from the same group are shown as mean ± SD; ^b^ *p* < 0.05, significant difference compared with the control group; ^c^ *p* < 0.05, significant difference compared with 10 TCID_50_ immune group; ^d^ *p* < 0.05, significant difference compared with 100 TCID_50_ immune group.

## Data Availability

The data generated during the study are openly available at https://www.scidb.cn/en/anonymous/UnJRbmV5 (accessed on 25 June 2024).

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
