# Peer review of "A Live Attenuated H1N1 Influenza Vaccine Based on the Mutated M Gene"

_vaccines, 2024, doi:10.3390/vaccines12070725_

Round 1

Reviewer 1 Report

Comments and Suggestions for Authors

The manuscript describes construction and evaluation of a live attenuated influenza vaccines. The authors evaluated live attenuated influenza virus containing the mutated M gene, results showed that mutated PR8(mPR8) can induce strong humoral and cellular immune responses, not only able to counter the challenge of homologous PR8 virus, but also provided cross-protection against heterologous H9N2 virus. These results were very attractive.

Major Comments:

1.    The authors claimed that mutated mPR8 provided cross-protection against heterologous H9N2 virus was due to non-neutralizing antibodies. The author should provide more evidence, e.g. neutralization antibodies test results before and after challenge.  

2.    According to the Figure 1B, mPR8 replicate in mouse bronchial alveolar, although titers decrease 1000 times, how long did the virus exist after inoculation?  Did the reisolated virus show any reverse mutation in M gene?

Minor comment:

3.    Line 371: Table1 and Line 380: Table2 should be in Results.

Author Response

We thank the reviewer for a very careful review and very thoughtful comments. We have made extensive modifications to our manuscript and supplemented extra data to make our results convincing, according to the editor and reviewers’ comments. In this revised version, changes to our manuscript were all highlighted within the document by using red colored text. The detailed point-by-point responses are listed below.

Reviewer#1: Comments and Suggestions for Authors

The manuscript describes construction and evaluation of a live attenuated influenza vaccines. The authors evaluated live attenuated influenza virus containing the mutated M gene, results showed that mutated PR8(mPR8) can induce strong humoral and cellular immune responses, not only able to counter the challenge of homologous PR8 virus, but also provided cross-protection against heterologous H9N2 virus. These results were very attractive.

We thank the reviewer for a very careful review and very thoughtful comments.

 Major Comments:

1. The authors claimed that mutated mPR8 provided cross-protection against heterologous H9N2 virus was due to non-neutralizing antibodies. The author should provide more evidence, e.g. neutralization antibodies test results before and after challenge.  

  We agree with the reviewer’s comment concerning this issue. A series of studies have shown that influenza virus hemagglutinin (HA) and neuraminidase (NA) can induce protective immunity in experimental animals as well as in humans. The antibodies (Abs) to the HA molecules neutralize the infectivity of the virus, while those to the NA efficiently prevent the release of the virus from the infected cells. The candidate vaccine strain used in this study was H1N1 (A/PR/8/34), isolated in 1934, while the challenging strain used to evaluate cross-protection was H9N2 subtype. The HA proteins of the two viruses share only 52.3% identity, while the NA proteins share only 44.75% identity. Due to the large antigenic difference between H1N1 and H9N2 viruses, we hypothesized that the cross-protection against H9N2 virus provided by the live attenuated vaccine in this study may not only depend on neutralizing antibodies, but it cannot be ruled out that antibodies may also play a certain role. Our previous studies showed that intranasal administration of recombinant nucleoprotein NP, M1, M2 could provide protection again different homologous or heterologous influenza viruses (1-3). In addition, in this study, although immunization of mice with LAIV (mPR8) developed by us provided 100% death cross- protection against from heterologous H9N2 virus challenge, but all mice had significant weight loss, indicating that immunized mice could still be infected with H9N2 virus. It was further demonstrated that LAIV did not stimulate the production of effective neutralizing antibodies against heterologous (H9N2) virus challenging in our study. In the following study, we will also conduct microneutralization analysis not only viral challenge experiment.

References:

1) Guo Lina, Zheng Mei, Ding Yahong, Li Dongmei, Yang Zhongdong, Wang Haiming, Chen Quanjiao, Sui Zhiwei, Fang Fang and Chen Z*. Protection against multiple influenza A subtypes by intranasal administration of recombinant nucleoprotein. Arch Virol. 2010 Nov;155(11):1765-75.

2) Sui Z, Chen Q, Fang F, Zheng M, Chen Z*. Cross-protection against influenza virus infection by intranasal administration of M1-based vaccine with chitosan as an adjuvant. Vaccine. 2010 Nov 10;28(48):7690-8.

3) Sui Z, Chen Q, Wu R, Zhang H, Zheng M, Wang H, Chen Z*. Cross-protection against influenza virus infection by intranasal administration of M2-based vaccine with chitosan as an adjuvant. Arch Virol. 2010 Apr;155(4):535-44

2. According to the Figure 1B, mPR8 replicate in mouse bronchial alveolar, although titers decrease 1000 times, how long did the virus exist after inoculation?  Did the reisolated virus show any reverse mutation in M gene?

  We thank the reviewer for the professional comment. We did not examine the duration of viral shedding in the respiratory tract of inoculated mice. We didn’t sequence the reisolated virus. We agree with the reviewer’s comment concerning these two issues. However, the mPR8 virus strain was serially passaged on MDCK cells, the 1st, 5th, and 10th generation viruses were harvested, and the viral RNA was extracted and sequenced to validate the nucleotide sequence of the virus. Sequence alignment suggested that the M gene of the mPR8 virus strain did not change from generations 1 to 10, indicating that the rescued strain has good genetic stability. The duration of viral shedding and the reverse mutation of the vaccine candidate strain in animal model are two major safety concerns of the live attenuated influenza vaccine. Since the major purpose of this study is to test whether the strategy or concept of constructing a live attenuated influenza vaccine based on mutated M gene is feasible, thus, these safety evaluations will be the focus of our future work.

Minor comment:

3. Line 371: Table1 and Line 380: Table2 should be in Results.

We thank the reviewer for pointing this out. This is a typographical error generated by the manuscript submission system. According to the reviewer's suggestion, we reformatted the manuscript. Tables 1 and 2 was in line 206.

Reviewer 2 Report

Comments and Suggestions for Authors

Line 35: do you mean influenza season?

This is a nice study with a good methodological basis and discussion section. I have no major comments other than the following minor points.

Line 42: do you mean epidemics and pandemics?

Line 50: suggest you include a reference to the beneficial nonspecific effects against other pathogens observed with live attenuated vaccines in general.

Line 57: do you mean budding or binding?

Section 2.8: why in bold?

Author Response

Thank you very much for a very careful review and very thoughtful comments. We have made extensive modifications to our manuscript and supplemented extra data to make our results convincing, according to the editor and reviewers’ comments. In this revised version, changes to our manuscript were all highlighted within the document by using red colored text. The detailed point-by-point responses are listed below.

 Comments and Suggestions for Authors

1. This is a nice study with a good methodological basis and discussion section. I have no major comments other than the following minor points.

We would like to thank the reviewers for their recognition and praise of our work.

2. Line 35: do you mean influenza season?

We thank the reviewer for pointing this out. We mean influenza season here. This sentence has been corrected to “The 2022-2023 influenza endemic in the UK has been more severe than in the previous two years”.

3. Line 42: do you mean epidemics and pandemics?

We thank the reviewer for pointing this out. We mean both epidemics and pandemics here. This sentence has been corrected to “…making it difficult to provide a sufficiently effective vaccine for use in the early stages of an influenza virus epidemics and pandemics.”.

4. Line 50: suggest you include a reference to the beneficial nonspecific effects against other pathogens observed with live attenuated vaccines in general.

We agree with the reviewer’s comment and include a reference (Peter Aaby, Mihai G Netea, Christine S Benn. Beneficial non-specific effects of live vaccines against COVID-19 and other unrelated infections. Lancet Infect Dis. 2023 Jan;23(1): e34-e42.) to the beneficial nonspecific effects against other pathogens observed with live attenuated vaccines in general. Here, we add the following sentence: “Moreover, live attenuated vaccines may have beneficial non-specific effects in preventing vaccine-unrelated infections.”.

5. Line 57: do you mean budding or binding?

We thank the reviewer for pointing this out. According to the reference listed here and other published literature, M2 initially stabilizes the site of influenza virus budding, and plays an important role in the formation of filamentous virions. So, we mean budding here.

6. Section 2.8: why in bold?

We thank you very much for pointing out this error. This is a typographical error. We have corrected this error in the latest version of the manuscript.

Reviewer 3 Report

Comments and Suggestions for Authors

The authors have submitted the manuscript titled "Live attenuated H1N1 influenza vaccine based on mutated M gene". In this, the authors tested the ability of the PR8 virus with M gene mutations as the live attenuated influenza vaccine (LAIV) in the mouse model.

I have described my concerns about the study below:

1. The authors keep switching between the terms 'Flu' and 'Influenza'. This must be kept consistent throughout the manuscript.

2. In lines 35-36 of the introduction, the authors talk about the 2022-2023 Influenza pandemic. However, there is no reported influenza pandemic. Do they mean endemic?

3. In most of the experiments, the authors have only 3 mice per group. Such a small number of animals is insufficient to draw any statistically significant conclusions.

4. Even though the negative control (PBS) group of mice is shown in the results section, the authors do not mention this group in the method section. This should be corrected.

5. The authors have a positive control in IFN-y secreting T cells experiment. Other than that, a positive control group is not included anywhere. The results of the mPR8 vaccine should be compared to the positive control group of animals to draw any conclusions related to vaccine effectiveness.

6. In the IFN-y secreting T cells experiment, even though the authors show a positive control group, it is unclear what the positive control used here is.

7. Tables 1 and 2 are wrongly placed. These should have been placed before Table 3. 

8. The results of the humoral immune response (IgG, IgA, IgG1, and IgG2a) should be graphically represented. These should have the baseline results from all the groups and a positive control group of animals.

9. The graphs showing IFN-y secreting T cells, CD4+, and CD8+ T cells should have a statistical comparisons and a positive control group of animals.

Comments on the Quality of English Language

The manuscript has some typos throughout. Those should be corrected and proofread. Other than that, the English language is fine.

Author Response

Thank you very much for a very careful review and very thoughtful comments. We have made extensive modifications to our manuscript and supplemented extra data to make our results convincing, according to the editor and reviewers’ comments. In this revised version, changes to our manuscript were all highlighted within the document by using red colored text. The detailed point-by-point responses are listed below.

1. The authors keep switching between the terms 'Flu' and 'Influenza'. This must be kept consistent throughout the manuscript.

We thank you very much for pointing out this error. We have corrected this error in the latest version of the manuscript.

2. In lines 35-36 of the introduction, the authors talk about the 2022-2023 Influenza pandemic. However, there is no reported influenza pandemic. Do they mean endemic?

We thank you very much for pointing out this error. We have corrected this sentence to: “The 2022-2023 influenza endemic in the UK has been more severe than in the previous two years”.

3. In most of the experiments, the authors have only 3 mice per group. Such a small number of animals is insufficient to draw any statistically significant conclusions.

We feel great thanks for your professional comments on our study. To evaluate the protective effect of rescued virus, 29 female BALB/c mice, aged 6-8 weeks, in each group were intranasally inoculated with different doses of mPR8 virus. At 21 days after immunization, 3 animals were randomly selected from each group, euthanized, and serum and spleen cells were isolated. The remaining 26 mice in each group were equally divided and challenged with 10× LD50 of PR8 or H9N2 influenza virus, respectively. Three days after virus challenge, three mice from each group were randomly selected and euthanized to measure virus titers in bronchial alveolar lavage fluid. The remaining mice (10 mice in each group) were used to observe body weight change and survival. When using mice as animal models to evaluate the protective effect of influenza vaccine challenge, survival and weight loss are generally considered to be the two most important indicators by influenza vaccine researchers. In this study, in all experiments involving mouse survival including “Pathogenicity determination in mice” and “Animal challenging experiment”, we set up 10 mice in each group. In other animal experiments in this study, we used 3 mice per group due to the concerns of animal welfare. All animal experiments in this study were repeated at least twice to verify the reliability and stability of the results.

4. Even though the negative control (PBS) group of mice is shown in the results section, the authors do not mention this group in the method section. This should be corrected.

We thank you very much for pointing out this error. We have added this information in the manuscript.

5. The authors have a positive control in IFN-y secreting T cells experiment. Other than that, a positive control group is not included anywhere. The results of the mPR8 vaccine should be compared to the positive control group of animals to draw any conclusions related to vaccine effectiveness.

We feel great thanks for your comments on our study. We set up positive and negative control wells in the IFN-γ-secreted T cell experiment for quality control and troubleshooting of this experiment. Since the major purpose of this study is to test whether the strategy or concept of constructing a live attenuated influenza vaccine based on mutated M gene is feasible, we did not use positive control vaccines, including live attenuated influenza vaccine and inactivated influenza vaccine. In further studies, we will compare the crossly protective ability of this M gene based attenuated vaccine with inactivated vaccines and other live attenuated vaccines, such as cold adaptive live attenuated vaccine.

6. In the IFN-y secreting T cells experiment, even though the authors show a positive control group, it is unclear what the positive control used here is.

We thank you very much for pointing out this omission. The isolated splenocytes cultured in complete culture medium consists of RPMI 1640 with the addition of 10% fetal bovine serum (FBS), ionomycin (50 ng/ml, InvivoGen) and phorbol myristate acetate (PMA, 10ng/ml, InvivoGen) was used as positive control. The isolated splenocytes cultured in complete culture medium with the addition of 10% FBS and PBS was used as negative control. We added this information in the manuscript.

7. Tables 1 and 2 are wrongly placed. These should have been placed before Table 3. 

We thank you very much for pointing out this error. This is a typographical error generated by the manuscript submission system. We have corrected this error in the latest version of the manuscript.

8. The results of the humoral immune response (IgG, IgA, IgG1, and IgG2a) should be graphically represented. These should have the baseline results from all the groups and a positive control group of animals.

Thanks for your suggestions. We have already described the antibody response by line chart with multiple dilution in Supplementary Figure 1. We also provide and described the figure here. In addition, in this study Ab-positive cutoff values were set as means â•‹ 2 × SD of negative control sera. An ELISA Ab titer was expressed as the highest serum dilution yielded a cutoff value >2 modified from the referenced papers lusted here.

References:

1) Sui Z, Chen Q, Fang F, Zheng M, Chen Z*. Cross-protection against influenza virus infection by intranasal administration of M1-based vaccine with chitosan as an adjuvant. Vaccine. 2010 Nov 10;28(48):7690-8.

2) Markus H Kainulainen, Eric Bergeron, Payel Chatterjee, Asheley P Chapman, Joo Lee, et al. High-throughput quantitation of SARS-CoV-2 antibodies in a single-dilution homogeneous assay. Sci Rep. 2021 Jun 10;11(1):12330. doi: 10.1038/s41598-021-91300-5.

3) Man Xing, Gaowei Hu, Xiang Wang, Yihan Wang, Furong He et al. An intranasal combination vaccine induces systemic and mucosal immunity against COVID-19 and influenza. NPJ Vaccines. 2024 Mar 21;9(1):64. doi: 10.1038/s41541-024-00857-5.

Supplementary Figure 1. Antibody responses in mice induced by intranasal administration of various doses of mPR8 vaccine. Formalin-inactivated whole PR8 virus vaccine was coated in 96-well ELISA plate. Then the serum sample to be tested was performed 2-times serial dilution firstly and then added on the coated plate. Biotinylated goat anti-mouse IgG antibody (Southern Biotechnology Associates, Inc. USA) was added. Then streptavidin-conjugated alkaline phosphatase (Southern Biotechnology Associates, Inc. USA) was added. After termination of the reaction, the OD450 absorbance was read using a SpectraMax M2e multifunction microplate reader (Molecular Devices). BALB/c mice were intranasally (i.n) immunized with different dose of mPR8 virus as indicated. The mice in control group were inoculated with the same volume of PBS, 3 mice per group. 21 days after immunization, the serum and nasal lavage fluid were collected for virus specific IgG(A) and IgA(B) assay respectively by ELISA. IgG1(C) and IgG(2a) in serum of the immunized mice were also determined by ELISA.

9. The graphs showing IFN-y secreting T cells, CD4+, and CD8+ T cells should have a statistical comparisons and a positive control group of animals.

We thank the reviewer for their comment. We have provided statistical comparisons in the latest version of the manuscript. We provided the modified figure here. As described in our response to Comment 5,we only set PBS control group in this study, we did not compare ability to provide protection of M gene based attenuated vaccine with other kinds of vaccine.

 Comments on the Quality of English Language

The manuscript has some typos throughout. Those should be corrected and proofread. Other than that, the English language is fine.

Round 2

Reviewer 3 Report

Comments and Suggestions for Authors

The authors have submitted the revised version of the manuscript "Live attenuated H1N1 influenza vaccine based on mutated M gene".

As pointed out in my previous comments as well, the authors need a positive control group in the experiments. The authors mentioned that they did not use a positive control because the major purpose of the study was to test the feasibility of constructing influenza vaccine with mutated M gene. However, this no conclusion regarding this can be drawn with a missing positive control in every experiment. To comment anything on the vaccine effectiveness, the experimental vaccine should be compared to a licenced influenza vaccine.

Other than this all my other concerns have been addressed appropriately.

Author Response

Comment: As pointed out in my previous comments as well, the authors need a positive control group in the experiments. The authors mentioned that they did not use a positive control because the major purpose of the study was to test the feasibility of constructing influenza vaccine with mutated M gene. However, this no conclusion regarding this can be drawn with a missing positive control in every experiment. To comment anything on the vaccine effectiveness, the experimental vaccine should be compared to a licensed influenza vaccine. Other than this all my other concerns have been addressed appropriately. Other than this all my other concerns have been addressed appropriately.

Response:

  Thank you very much for your comments. In this study, the parental strain of the live attenuated influenza vaccine candidate is A/Puerto Rico/8/34 (H1N1, PR8), one of first influenza virus strains isolated from humans in 1934, which is the classic influenza virus model strain used by most laboratories engaged in influenza research around the world. The isolated PR8 isn’t lethal to mouse at first. After serial passage in the lung of mice, the pathogenicity of the strain was enhanced, showing lethality to mice, and multiple amino acid mutations occurred in its genome. The strain obtained after adaptation through serial mouse lung passages is also called mouse-adapted strain.

  Our published results also demonstrated that influenza virus needs to undergo multiple passages in the mouse lung before it can effectively infect mice and exhibit obvious pathogenicity and lethality (see the reference below). Our published results also showed that there were multiple amino acid mutations (22 amino acid mutations) occurred in the genome of influenza virus during passage in mouse lung. Most licensed live attenuated influenza vaccines (LAIV) have been developed for humans or livestock, they may not infect mice or replicate effectively in mice, which means it can’t stimulate the immune response effectively in mouse. Although the LAIV strain can be adapted and acquire the ability to replicate efficiently in mice through serial mouse lung passage, many unpredictable or uncertain mutations will occur in this process, such as pathogenicity and virulence recovery and loss of its genetic characteristics as a live attenuated vaccine. Therefore, the licensed LAIV is not a good positive control in our study based on mouse model.

  On the other hand, the inactivated influenza vaccine can’t be used as a positive control. Inactivated vaccines are mostly administered through intramuscular injection to activate humoral immunity, while live attenuated vaccines are mostly administered through nasal route to stimulate both humoral and cellular immunity. The protective mechanisms of the two vaccines are different.

  In addition,a licensed influenza vaccine was prepared by viral strains recommend by WHO. WHO convenes technical consultations in February and September each year to recommend viruses for inclusion in influenza vaccines for the northern hemisphere and southern hemisphere influenza seasons, respectively. The WHO recommendation on vaccine composition is based on the year-round work of the WHO Global Influenza Surveillance and Response System (GISRS). It is based on global influenza activity. There is no licensed influenza vaccine based on PR8 strain.

  From what has been discussed above, it’s difficult to select one suitable licensed vaccine as positive control in this study. However, we believe that in the study of LAIV, how to establish an appropriate positive control is a problem that needs to be seriously considered. In further studies, we will compare the protective ability of this M gene based attenuated vaccine with other live attenuated vaccines, such as cold adaptive live attenuated vaccine in mouse models.

Reference:

1. Hongbo Zhang, Bing Xu, Quanjiao Chen, Jianjun Chen, Ze Chen. Characterization of an H10N8 influenza virus isolated from Dongting lake wetland. Virol J. 2011 Jan 27:8:42. doi: 10.1186/1743-422X-8-42.